# Lower termite (*Coptotermes heimi*) gut fibrolytic bacterial consortium: Isolation, phylogenetic characterization, fibre degradation potential and *in vitro* digestibility

**Pankaj Kumar Kumawat[1,2], Srobana Sarkar[1]\*, Satish Kumar[3], Artabandhu Sahoo[1,4]\***

**1** ICAR- Central Sheep and Wool Research Institute, Mewar University, Avikanagar, Rajasthan, India, **2** Mewar University, Chittorgarh, Rajasthan, India, **3** Mata Basanti Devi School of Biosciences & Biotechnology, Agra, India, **4** ICAR- National Research Centre on Camel, Bikaner, Rajasthan, India

\* sarkarsrobana@gmail.com (SS); sahooarta1@gmail.com (AS)

## Abstract

Lower termites produce wide array of fibrolytic enzymes and serves as prospective microbial enzymes source for enhancing biodegradability of recalcitrant ligno-cellulosic fibrous feeds. The present study was aimed to isolate and characterize anaerobic fibrolytic bacteria from gut of termite *Coptotermes heimi* for screening promising isolates to improve fiber digestibility in ruminants. A total of 141 isolates were obtained from 97 termite gut samples, and 24 isolates (TM1 to TM24) were selected and characterized as fibrolytic. All isolates were obligatory anaerobes and catalase negative except, TM8, TM9, TM14 and TM22 which were facultative anaerobes and catalase positive. Overall fibrolytic enzyme activity was highest in isolate TM23, TM6 and TM22. Highest FPase activity was observed in isolate TM5 (12.05 U/ml) while, lowest in TM19 (6.41 U/ml). The phylogenetic analysis of the isolates depicted four major families, i.e., *Clostridiales, Bacillales, Lactobacillales* and *Enterobacterales* under phyla *Firmicutes* and *Proteobacteria*. The *in vitro* dry matter digestibility of the substrate was increased by 9.4 to 36.0% with the inoculation of isolated bacterial strains. Among the screened isolates, TM6 exhibited highest ability to improve the *in vitro* dry matter digestibility. The findings of the present study revealed that the fibrolytic bacteria isolated from - termite gut can be used for commercial enzyme production or in rumen biotechnological application for enhancing utilization of fibrous feed in ruminants.

## Introduction

Termites act as "ecosystem engineers" because they help in efficient degradation of complex plant biomass into resourceful bioenergy in comparison to other invertebrates [1, 2]. Their potential to degrade fibrous polysaccharide depends on the dual enzymatic machinery of termite gut and its gut microbial symbionts, which makes them an ideal source of microbial enzymes [3, 4]. Termites have the ability to degrade around 65 to 87% hemicelluloses and 74 to 99% celluloses [5]. They are intricately grouped into higher and lower termites [6], the higher termites harbour only prokaryotes and lacks protists [5,7,8], while the lower termites

**Data availability statement:** All relevant data are within the manuscript.

**Funding:** The author(s) received no specific funding for this work.

**Competing interests:** The authors have declared that no competing interests exist.

contain both protists and bacteria in their gut [9]. The microbes present in termite gut work in consortium and help in breaking down complex carbohydrates [10, 11]. However, the abundance of microbes inhabiting in the gut of termites not only depends on their hierarchy but also on their geographical location and available vegetation [12].

Utilization of agro-industrial by-products and poor-quality fibrous feeds in animal nutrition plays a pivotal role in sustainable agriculture. Effect of several treatments including physical, chemical and biological have been studied for enhancing the utility of poor-quality feed resources in ruminants [13, 14]. In ruminants, studies have shown that exogenous fibrolytic enzymes aid in increasing the digestibility of fiber [15]. Earlier studies reported that bacteria isolated from termite gut have the ability to disintegrate lignin barrier and digest carbohydrate polymers [13,16–18]. Additionally, because the environment- in the rumen and termites' gut- are similar in terms of anaerobiosis, reductive acetogenesis, methanogenesis, and production of volatile fatty acids, thus the bacteria isolated from termite guts are better suited as microbial feed additives to enhance the utilization of recalcitrant ligno-cellulosic fibrous crop residues. [16]. *Coptotermes heimi* are the most predominant lower termites the in semi-arid regions of India [19]. Till date, little information is available regarding origin and evolution of *Coptotermes heimi* and no research has been done on their gut microbiota's abundance. It is hypothesized that *Coptotermes heimi* harbour a diverse species of fiber degrading bacteria with promising fibrolytic enzyme potential. Therefore, the present study was undertaken to isolate and identify novel fibrolytic bacteria from the gut of *Coptotermes heimi* and to evaluate their capacity to break down plant fibre in terms of enzymatic activity and potential to improve digestibility of roughage-based diets.

## Materials and methods

### Specimen collection

The present study was conducted at and surrounding areas of ICAR-Central Sheep and Wool Research Institute campus, Rajasthan, India. The termite samples of *Coptotermes heimi* were collected from nests on *Ailanthus excelsa*, *Azadirachta indica*, *Albizia lebbeck*, *Cassia fistula*, *Mangifera indica* and *Eucalyptus camaldulensis* trees. A total of ninety-seven samples of wood eating termites were collected (in March 2019) and transported to the laboratory and washed with water to remove the dirt. Thereafter, the termite samples were washed with 70% ethanol followed by distilled water for surface sterilization. The termite's head was removed using sterilized needle and the entire gut was excised out [20]. The gut samples were macerated and processed for isolation of fibre degrading bacteria.

### Isolation of fibre degrading bacteria

For isolation of fiber degrading bacteria, M10 medium was used with slight modification which comprised of 15 ml mineral solution-I (0.3% $K_2HPO_4$), 15 ml mineral solution –II (0.3% $KH_2PO_4$, 0.6% $(NH_4)_2SO_4$, 0.6% NaCl, 0.06% $MgSO_4.7H_2O$, 0.0795% $CaCl_2.2H_2O$), 0.05% tryptone, 0.05% yeast extract powder, 0.1% resazurine, 0.2% microcrystalline cellulose, 0.1 ml Pfennig trace elements solution, 0.31 ml volatile fatty acid mixture, 0.8% sodium bicarbonate ($NaHCO_3$), 20 ml clarified rumen liquor, 0.05% cystine hydrochloride (Cys-HCL). The collected samples were serially diluted in anaerobic diluent [21] up to $10^8$ cells/ml. From each dilution, around 0.5 ml of inoculum was inoculated in M10 agar medium (1.5% agar) in Hungate tubes (5 mL) using disposable syringe. The agar tubes were rolled on ice bed for solidification and incubated at 39°C for 48 h. Presumptive colonies of different colour, size and shape were picked and inoculated in anaerobically prepared M10 broth medium for sub-culturing in an anaerobic chamber (Bioxia, India). The above process was repeated at least for 4 times till

monoculture was obtained. The pure cultures isolated were stored in 80% anaerobic glycerol stock at -80°C for further use.

## Morphological and biochemical characterization of isolates

The isolates were characterized morphologically by Gram staining [22] and biochemically by tests like anaerobiosis, catalase, gas production, motility, $H_2S$ production, indole, methyl red and sugar fermentation [23]. The results obtained were compared with Bergey's Manual of Determinative Bacteria for identification [24].

## Fibrolytic enzyme activity

The experiment was conducted to estimate the fibrolytic enzymatic activity, 96 h old bacterial culture, in 3 replicates, were treated with lysozyme and sonicated for 6 min with 30-sec pulse rate under ice-cool condition and centrifuged at $14,000 \times g$ [23]. The supernatant obtained was used as a crude enzyme extract for estimation of fibrolytic enzymes like carboxymethyl cellulase (CMCase, endoglucanase), avicelase (exoglucanase), α-amylase (1,4-Gluanohydrolase), xylanase (Endo-1,4-β-xylanase), β-glucosidase and filter paper degrading assay (FPase) [25,26]. The endoglucanase, exoglucanase, FPase, and α-amylase enzyme activities were expressed as μmol glucose released per min per ml (U/ml) and xylanase activity was expressed as μmole xylose released per min per mL (U/mL). The substrate used for estimation of endoglucanase, exoglucanase, α-Amylase, and xylanase were 1% carboxymethyl cellulose (Sigma-Aldrich, USA), 1% avicel (Sigma-Aldrich, USA), 1% starch (Sigma-Aldrich, USA), and 1% xylan (Sigma-Aldrich, USA), respectively while, the FPase activity was determined using 50 mg Whatman filter paper as the substrate. To calculate the activity of endoglucanase, exoglucanase, α-Amylase, and FPase enzyme, 0.1% glucose (Sigma-Aldrich, USA) was used as a standard whereas, for xylanase enzyme 0.1% xylose was used as a standard (Sigma-Aldrich, USA). The test reaction consisted of 0.05 ml enzyme extract, 0.05 ml substrate, and 0.1 ml phosphate buffer. The control reaction was identical to the test with the exception of the enzyme extract, which was denatured in the control reaction. The contents in the tubes were mixed properly and kept at 39°C for 60 min. Following incubation, 0.5 ml di-nitro salicylic acid reagent (Sigma-Aldrich, USA) was added to the tubes and kept in boiling water bath for 10 min. After 10 min, 0.1 ml of Rochelle salt was added to the tubes and allowed to cool to room temperature and the absorbance was measured at 575 nm against blank. For the determination of β-glucosidase enzyme activity, 0.9 ml of 0.1% PNPG (p-nitrophenol β-D-glucopyranose) (Hi-media, India) was incubated for 10 min with 0.1 ml of enzyme extract as the substrate. The control tubes contained 0.1 ml denatured enzyme extract as the substrate. Following incubation, 1.0 ml of 2% sodium carbonate was added to the tubes to stop the reaction and absorbance was measured at 400 nm against blank. For the preparation of standard curve, 0.002% p-nitrophenol was used (Sigma-Aldrich) and the enzyme activity was expressed as μmole p-nitrophenol released per min per ml.

## Filter paper degradation potential

For assessing crystalline cellulose degrading potential of the isolates, M10 medium was prepared anaerobically and $25 \times 5$ mm strip of filter paper (Whatman No. 1) was added in the media and autoclaved. The isolates were inoculated in roll tubes containing filter paper media and incubated at 39°C for 10 d. Degradation of crystalline cellulose was determined as the amount of filter paper degraded at an interval of 24 h up to 10 d till complete degradation. Degradation potential of test cultures was measured after comparison with un-inoculated control samples [23].

## Molecular characterization and sequence homology analysis of 16S rRNA gene

Genomic DNA was extracted from the isolates by using DNeasy Power Lyzer Power Soil Kit (Qiagen, Germany) as per manufacturer's instructions. The 16S rRNA gene was amplified from the isolated genomic DNA with 27F 5'-AGAGTTTGATCCTGGCTCAG-3' (forward) and 1495R 5' CTACGGCTACCTTGTTACG-3' (reverse) universal 16S rRNA gene primers. A polymerase chain reaction (PCR) was carried out in an Eppendorf thermal cycler (Germany), each reaction mixture (50 μl) consisted of 10X reaction buffer, 10 μM primers, 10 mM dNTPs, 1.5 U Taq DNA polymerase (Thermo Scientific, Germany), and up to 100 ng DNA template. The annealing temperature was 52°C and the PCR product size was 1.5 kb. The quality of amplified product was checked by agarose gel electrophoresis. The amplified products were gel purified using GeneJET Gel extraction kit (Cat#K0691, Thermo Scientific, Germany) as per manufacturer's protocol. The purified amplified product was sequenced as per sanger sequencing method by an outsourcing agency (Eurofins, Bengaluru). The nucleotide sequences obtained were trimmed using edit seq software (DNA STAR) and analysed for nearest valid taxon and species using BLAST algorithm [27]. All the aligned sequences were submitted to GenBank. The evolutionary history was inferred using the Neighbor-Joining method. The tree was drawn to scale with branch lengths (above the branches) in the same units as those of the evolutionary distances used to infer the phylogenetic tree. The evolutionary distances were computed using the Maximum Composite Likelihood method and were in the units of the number of base substitutions per site and the analysis involved 40 nucleotide sequences. All ambiguous positions were removed for each sequence pair (pairwise deletion option). There were a total of 1605 positions in the final dataset and evolutionary analysis was conducted in MEGA11.

### *In vitro* true digestibility

To evaluate the digestibility potential of the screened isolates an *in vitro* gas production test was conducted [28, 29]. Around 500 ± 10 mg of dried *Cenchrus ciliaris* and *Vigna mungo* straw (80:20) was used as a substrate. Both live culture and autoclaved cultures (control) were used for each isolate. Each treatment had 4 replications. Around 40 ml incubation medium was administered anaerobically in a 100 ml glass syringe along with 3 ml of culture. After 48 h incubation at 39°C, the contents of the syringe were refluxed with 100 ml of neutral detergent solution (sodium lauryl sulphate - 30 g, disodium ethylene diamino tetra acetate - 18.61 g, sodium borate decahydrate - 6.81 g, disodium hydrogen phosphate (anhydrous) - 4.56 g, triethyleneglycol − 10 ml, distilled water - 990 ml per L) in a spoutless beaker, filtered through pre-weighed gooch crucibles and weighed after 24 h oven drying to determine *in vitro* dry matter digestibility [30].

## Short chain fatty acid production

After conducting *in vitro* gas production test as mentioned above, the content of the syringe was strained with muslin cloth, collected in scintillation vials and preserved at -20°C by adding 0.25% m-phosphoric acid (Hi-media) in the ratio of 9:1. For estimation of short chain fatty acids (SCFAs), the preserved samples were centrifuged at 5000 × g for 10 min and the clear supernatant obtained was injected (10 μl) into the gas chromatogram (GC). The GC (Dani Make, GC-1000, Italy) was equipped with a stainless-steel column of 6 ft length and 1/8 inch diameter, packed with 10% SP-1200 and 1% phosphoric acid (Supelco). The temperature of the injector and detector was 220°C and 250°C, respectively while, the oven temperature was standardized at a gradient of 115°C to 150°C for separation of each molecule. Nitrogen

gas was used as a carrier with a pressure of 0.8 bar while, the pressure of air and hydrogen was 1.0 and 0.8 bar, respectively. The standard used was prepared by mixing 65, 20, 3, 8, 2 and 2 mM of acetic acid, propionic acid, iso-butyric acid, butyric acid, iso-valeric acid and valeric acid, respectively.

## Statistical analysis

The data obtained were analyzed using one-way analysis of variance (ANOVA). Tukey's b test was used for the post-hoc comparison of different treatment means and statistical differences were considered significant at $P < 0.05$.

## Results

### Morphological and biochemical characterization

Out of 141 isolates, 24 (designated as TM1- TM24) were selected on the basis of formation of clear zones on Congo red plates for further microscopic and biochemical characterization (Table 1). All the isolates were tested for anaerobiosis and catalase reaction for assessing their ability to tolerate anoxic environment. Twenty of the 24 isolates showed positive anaerobiosis and negative catalase reaction, indicating that they are obligatory anaerobes while, the other 4 isolates were facultative anaerobes with positive catalase reactions. Except for TM12, all the isolates

**Table 1. Morphological and biochemical characteristics of different isolates.**

| Isolates | Gram Stain | Shape | Gas Production Test | Anaerobiosis Test | Catalase Test | Motility Test | $H_2S$ Production Test | Indole Production Test | Methyl Red Test |
|---|---|---|---|---|---|---|---|---|---|
| TM1 | + | Rod | – | + | – | Motile | + | – | – |
| TM2 | – | Rod | + | + | – | Motile | + | – | + |
| TM3 | – | Rod | + | + | – | Motile | + | – | + |
| TM4 | – | Rod | + | + | – | Motile | + | – | + |
| TM5 | – | Rod | – | + | – | Motile | + | – | – |
| TM6 | + | Rod | – | + | – | Non motile | + | – | + |
| TM7 | + | Rod | – | + | – | Motile | + | – | – |
| TM8 | + | Rod | + | – | + | Non motile | + | – | + |
| TM9 | + | Rod | – | – | + | Motile | + | – | + |
| TM10 | + | Rod | – | + | – | Motile | + | – | + |
| TM11 | + | Rod | + | + | – | Non motile | + | – | – |
| TM12 | + | Rod | – | + | – | Motile | – | – | + |
| TM13 | + | Rod | – | + | – | Motile | + | – | + |
| TM14 | – | Rod | + | – | + | Non motile | + | – | + |
| TM15 | – | Rod | + | + | – | Non motile | + | – | + |
| TM16 | + | Cocci | + | + | – | Non motile | + | – | + |
| TM17 | + | Rod | – | + | – | Motile | + | – | + |
| TM18 | + | Cocci | + | + | – | Non motile | + | – | + |
| TM19 | + | Rod | – | + | – | Non motile | + | – | + |
| TM20 | + | Rod | – | + | – | Non motile | + | – | + |
| TM21 | – | Rod | + | + | – | Motile | + | – | + |
| TM22 | – | Rod | – | – | + | Non motile | + | – | + |
| TM23 | + | Rod | + | + | – | Non motile | + | – | + |
| TM24 | + | Rod | + | + | – | Motile | + | – | – |

(-) Non-utilizers, (+) Utilizers.

produced hydrogen sulfide gas by the converting sulfur to sulfide. None of the isolates produced indole from the reductive deamination of tryptophan; hence all the isolates were indole negative. All the isolates were able to utilize mannose, sucrose, glucose, carboxy-methyl cellulose, lactose, arabinose, starch, maltose, fructose, xylose and xylan for growth except ribose (Table 2).

## Fibrolytic enzyme activity

In this study, endoglucanase, exoglucanase, FPase, β-glucosidase, α-amylase, and xylanase enzyme assays were performed to evaluate the fibrolytic potential of the isolates (Table 3). The isolates showed varying degree of fibrolytic enzyme activity with different substrates like carboxymethyl cellulose, avicel, filter paper, p-nitrophenyl-β-glucoside, starch and xylan. The endoglucanase activity ranged from 4.83 to 7.24 U/ml and highest activity was shown by isolate TM23. The exoglucanase activity among the isolates ranged from 4.24 to 8.40 U/ ml and highest exoglucanase activity was showed by TM6 and lowest by TM19. The FPase activity was highest in isolate TM5 (12.05 U/ml) and, lowest in TM19 (6.41 U/ml). Maximum β-glucosidase activity was observed in TM18 and lowest in TM1. The α-amylase activity in the isolates ranged from 2.04 to 12.42 U/ml, where, highest and lowest activity was observed in isolate TM22 and TM3, respectively. Xylanase activity ranged from 0.055 to 12.00 U/ml, maximum being in TM5 and minimum in TM20.

**Table 2. Utilization of various carbohydrates by different isolates.**

| Isolates | 1 | 2 | 3 | 4 | 5 | 6 | 7 | 8 | 9 | 10 | 11 | 12 | 13 |
|---|---|---|---|---|---|---|---|---|---|---|---|---|---|
| TM1 | + | + | + | + | – | – | + | + | + | + | + | + | + |
| TM2 | + | + | + | + | + | + | + | + | + | + | + | + | + |
| TM3 | + | + | + | + | + | + | + | + | + | + | + | + | – |
| TM4 | + | + | + | + | + | + | + | + | + | + | + | + | – |
| TM5 | + | + | + | + | + | + | + | + | + | + | + | + | – |
| TM6 | + | – | + | + | + | + | + | + | + | + | + | + | – |
| TM7 | + | + | + | + | – | – | + | + | + | + | + | + | – |
| TM8 | + | + | + | + | + | + | + | + | + | + | + | + | – |
| TM9 | + | + | + | + | + | + | + | + | + | + | + | + | – |
| TM10 | – | + | + | + | + | – | + | + | + | + | + | + | + |
| TM11 | + | + | + | + | + | + | + | + | + | + | + | + | + |
| TM12 | + | – | + | + | + | + | + | + | + | + | + | + | + |
| TM13 | – | + | + | + | + | + | + | – | + | + | + | + | + |
| TM14 | + | + | + | + | + | + | + | – | + | + | + | + | + |
| TM15 | – | + | + | + | – | – | + | – | + | + | + | + | + |
| TM16 | + | + | + | + | + | + | + | + | + | + | + | + | – |
| TM17 | + | + | + | + | + | + | + | + | + | + | + | + | + |
| TM18 | – | + | – | + | + | + | + | – | + | + | + | + | – |
| TM19 | + | + | + | + | + | + | + | + | + | + | + | + | + |
| TM20 | – | + | + | + | + | – | + | – | + | + | + | + | – |
| TM21 | – | + | + | + | + | + | + | – | + | + | + | + | + |
| TM22 | + | + | + | + | + | + | + | + | + | + | + | + | – |
| TM23 | + | + | + | + | + | + | + | + | + | + | + | + | + |
| TM24 | + | + | – | + | + | + | + | – | + | + | + | + | + |

(-) Non-utilizers, (+) Utilizers.

1. Galactose, 2. Mannose, 3. Sucrose, 4. Glucose, 5. Xylose, 6. Xylan, 7. Carboxy-methyl cellulose, 8. Fructose, 9. Lactose, 10. Arabinose, 11. Starch, 12. Maltose, 13. Ribose.

**Table 3. Fibrolytic enzyme activities of bacterial isolates.**

| Isolates | 1 | 2 | 3 | 4 | 5 | 6 |
|---|---|---|---|---|---|---|
| TM1 | 5.21[abcd] | 7.08[efghi] | 8.58[bcdef] | 0.009[a] | 5.17[cdef] | 1.72[abcd] |
| TM2 | 5.00[ab] | 6.53[defghi] | 8.03[abcde] | 0.071[d] | 3.35[abc] | 8.44[kl] |
| TM3 | 5.69[abcdef] | 5.87[bcdef] | 8.93[cdef] | 0.015[ab] | 2.04[a] | 3.11[cdefg] |
| TM4 | 5.51[abcdef] | 6.16[cdef] | 8.45[bcdef] | 0.026[abc] | 4.16a[bcd] | 2.88[bcdef] |
| TM5 | 4.83[a] | 5.57[bcde] | 12.05[g] | 0.041[abcd] | 5.32c[defg] | 12.00[m] |
| TM6 | 7.60[h] | 8.40[j] | 8.67[bcdef] | 0.059[cd] | 7.34f[ghi] | 5.55[hij] |
| TM7 | 5.97[bcdefg] | 7.08[efghi] | 8.58[bcdef] | 0.009[a] | 5.17[cdef] | 1.72[abcd] |
| TM8 | 6.31[efg] | 7.64[ghij] | 8.35[bcdef] | 0.072[d] | 5.68[cdefgh] | 3.44[defg] |
| TM9 | 6.83[gh] | 7.49[fghij] | 9.64[ef] | 0.031[abcd] | 4.36[abcd] | 1.11[abc] |
| TM10 | 6.36[fg] | 8.14[ij] | 8.93[cdef] | 0.055[bcd] | 6.03[defgh] | 0.83[ab] |
| TM11 | 5.69[abcdef] | 6.22[cdefg] | 8.41[bcdef] | 0.040[abcd] | 2.19[ab] | 2.88[bcdef] |
| TM12 | 5.90[bcdefg] | 7.69[hij] | 7.81[abcd] | 0.371[fg] | 10.73[jk] | 5.11[ghij] |
| TM13 | 5.30[abcde] | 5.37[abcde] | 9.42[def] | 0.396[gh] | 5.02[cdef] | 2.11[abcde] |
| TM14 | 5.92[cdefg] | 6.38[cdefgh] | 8.80[bcdef] | 0.384[fgh] | 9.67[ij] | 10.16[lm] |
| TM15 | 5.37[bcdef] | 6.11[cdef] | 7.12[ab] | 0.407[gh] | 8.10[hi] | 0.66[a] |
| TM16 | 5.25[abcd] | 5.12[abc] | 8.76[bcdef] | 0.501[k] | 2.34[ab] | 5.83[ij] |
| TM17 | 6.21[defg] | 5.53[bcde] | 7.67[abc] | 0.454[ij] | 11.23[jk] | 3.66[defgh] |
| TM18 | 5.63[abcdef] | 5.55[bcde] | 7.44[abc] | 0.541[l] | 3.35[abc] | 5.00f[ghij] |
| TM19 | 5.65[abcdef] | 4.24[a] | 6.41[a] | 0.465[jk] | 3.76a[bcd] | 3.92[efghi] |
| TM20 | 5.17[abc] | 4.79[ab] | 7.52[abc] | 0.423[hi] | 7.85[ghi] | 0.55[a] |
| TM21 | 5.32[abcde] | 5.20[abcd] | 7.24[abc] | 0.349[f] | 4.69[bcde] | 7.1[jk] |
| TM22 | 6.28[efg] | 6.38[cdefgh] | 9.92[f] | 0.124[e] | 12.42[k] | 8.27[kl] |
| TM23 | 7.24[h] | 6.18[cdef] | 7.95[abcde] | 0.051[abcd] | 6.36[defgh] | 6.77[jk] |
| TM24 | 5.37[abcdef] | 6.64[defgh] | 8.40[bcdef] | 0.073[d] | 3.08[abc] | 10.22[lm] |
| SEM | 0.08398 | 0.12864 | 0.13664 | 0.0222 | 0.33731 | 0.37590 |
| pValue | 0.000 | 0.000 | 0.000 | 0.000 | 0.000 | 0.000 |

1. Endoglucanase Activity (μmol glucose/h/ml or units/ml), 2. Exoglucanase Activity (μmol glucose/h/ml or units/ml), 3. FPase Activity (μmol glucose/h/ml or units/ml), 4. β-Glucosidase Activity (μmol p-Nitrophenol/min/ml or units/ml), 5. α-Amylase Activity (μmol glucose/h/ml or units/ml), 6. Xylanase Activity (μmol xylose/h/ml or units/ml), [a-m] means bearing superscript differs significantly in a column (P < 0.05).

### Filter paper degradation potential

In the present study, filter paper degradation potential of the isolates was determined at regular intervals of 24 h up to 10 d following the inoculation of bacterial isolates in filter paper strip media (Table 4). Among the isolates, TM5, TM9 and TM13 degraded filter paper in 7 days compared to others and these isolates also possessed higher FPase activity compared to other isolates.

### Molecular identification and evolutionary relationships of isolates

The PCR product of ~ 1500 bp size obtained after 16S rRNA gene amplification (Fig 1) was considered for all the isolates for molecular identification (Table 5). The isolates TM1 and TM7 were 99% similar to *Bacillus megaterium* IAM 13418. The isolates TM2 and TM4 had 98% similarity to *Citrobacter freundii* JCM1657. The isolates TM10 and TM13 were 99% and 98% similar to *Clostridium termitidis,* respectively. The isolates TM16 and TM 18 were 97% and 99% similar to *Lactococcus nasutitermitis* M19, while, TM17 and TM24 showed 90% and 99% similarity with *Clostridium sporogenes* JCM 1416. The isolates TM19 and TM20 were 98% similar to *Bacillus circulans* ATCC 4513. Isolate TM3 was 99% similar with *Citrobacter*

**Table 4. Filter paper degradation potential of different isolates.**

| Isolates | 24h | 48h | 72h | 96h | 5D | 6D | 7D | 8D |
|---|---|---|---|---|---|---|---|---|
| TM1 | + | ++ | ++ | +++ | ++++ | +++++ | ++++++ | CD |
| TM2 | + | + | ++ | +++ | ++++ | +++++ | ++++++ | CD |
| TM3 | + | ++ | ++ | +++ | ++++ | +++++ | ++++++ | CD |
| TM4 | + | ++ | ++ | +++ | ++++ | +++++ | ++++++ | CD |
| TM5 | ++ | +++ | ++++ | +++++ | +++++ | ++++++ | CD | |
| TM6 | + | ++ | +++ | ++++ | +++++ | +++++ | ++++++ | CD |
| TM7 | + | ++ | ++ | +++ | ++++ | +++++ | ++++++ | CD |
| TM8 | + | + | ++ | +++ | ++++ | +++++ | ++++++ | CD |
| TM9 | + | ++ | +++ | ++++ | +++++ | ++++++ | CD | |
| TM10 | + | ++ | +++ | ++++ | +++++ | +++++ | ++++++ | CD |
| TM11 | + | ++ | ++ | +++ | ++++ | +++++ | ++++++ | CD |
| TM12 | + | + | ++ | +++ | ++++ | +++++ | ++++++ | CD |
| TM13 | + | ++ | +++ | ++++ | +++++ | ++++++ | CD | |
| TM14 | + | + | ++ | +++ | ++++ | +++++ | ++++++ | CD |
| TM15 | + | + | ++ | +++ | ++++ | +++++ | ++++++ | CD |
| TM16 | + | + | ++ | +++ | ++++ | +++++ | ++++++ | CD |
| TM17 | + | ++ | ++ | +++ | ++++ | +++++ | ++++++ | CD |
| TM18 | + | ++ | ++ | +++ | ++++ | +++++ | ++++++ | CD |
| TM19 | + | + | ++ | +++ | ++++ | +++++ | ++++++ | CD |
| TM20 | + | + | ++ | +++ | ++++ | +++++ | ++++++ | CD |
| TM21 | + | + | +++ | +++ | ++++ | +++++ | ++++++ | CD |
| TM22 | + | ++ | ++ | +++ | ++++ | +++++ | ++++++ | CD |
| TM23 | + | + | ++ | +++ | ++++ | +++++ | ++++++ | CD |
| TM24 | + | ++ | ++ | +++ | ++++ | +++++ | ++++++ | CD |

'+' = 25% degradation, '++' = 50% degradation, '+++' = 60% degradation, '++++' = 70% degradation, '+++++' = 80% degradation, '++++++' = 90% degradation, CD = Complete Degradation of filter paper (100%), h = Hours, D = Days.

*farmeri* CDC 2991, TM5 was 99% similar with *Bacillus flexus* Tl-1, TM8 was 99% similar to *Bacillus licheniformis* ATCC14580, TM9 was 98% similar to *Bacillus cereus* ATCC 14579, TM11 was 100% similar to *Clostridium saccharobutylicum* P262, TM12 was 97% similar to *Clostridium indolis* 7, TM14 was 98% similar to *Bacillus oleronius* DSM9356, TM15 was 96% similar to *Clostridium cellulovorans* 743B, TM21 was 98% similar to *Ruminiclostridium papyrosolvens* DSM 2782, TM22 was 96% similar to *Brevibacillus brevis* DSM 30 and TM23 was 96% similar to *Clostridium puniceum* BL 70. All the 16S rRNA gene sequences (TM1 - TM24) were submitted to NCBI GenBank and accession no. MT356127 to MT356150 were obtained.

### *In vitro* dry matter digestibility

As depicted in Fig 2, inoculation of bacterial isolates from termite gut enhanced (P < 0.05) the dry matter (DM) digestibility of substrate containing *Cenchrus ciliaris* hay and *Vigna mungo* straw by 9 to 36% in comparison to control. Inoculation of isolate TM6 resulted in maximum increase of DM digestibility than other bacterial isolates.

### Short chain fatty acid production

Total short chain fatty acid production by the isolates ranged from 64.19 mM to 117.73 mM (Table 6). The concentration of acetate, propionate, iso-butyrate, butyrate, iso-valerate and

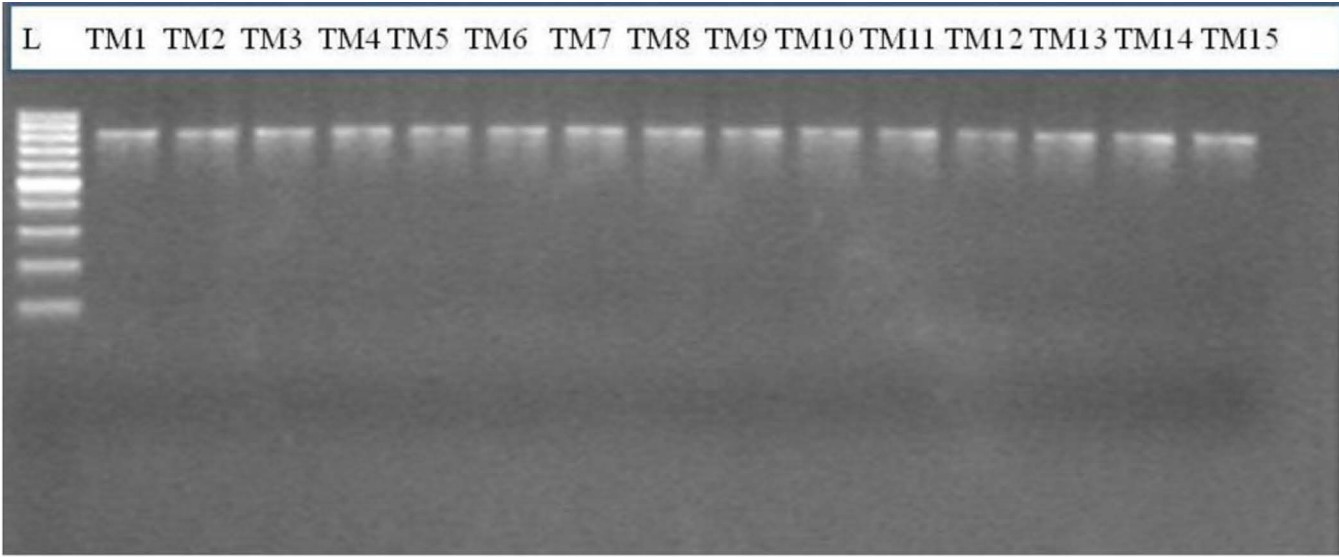

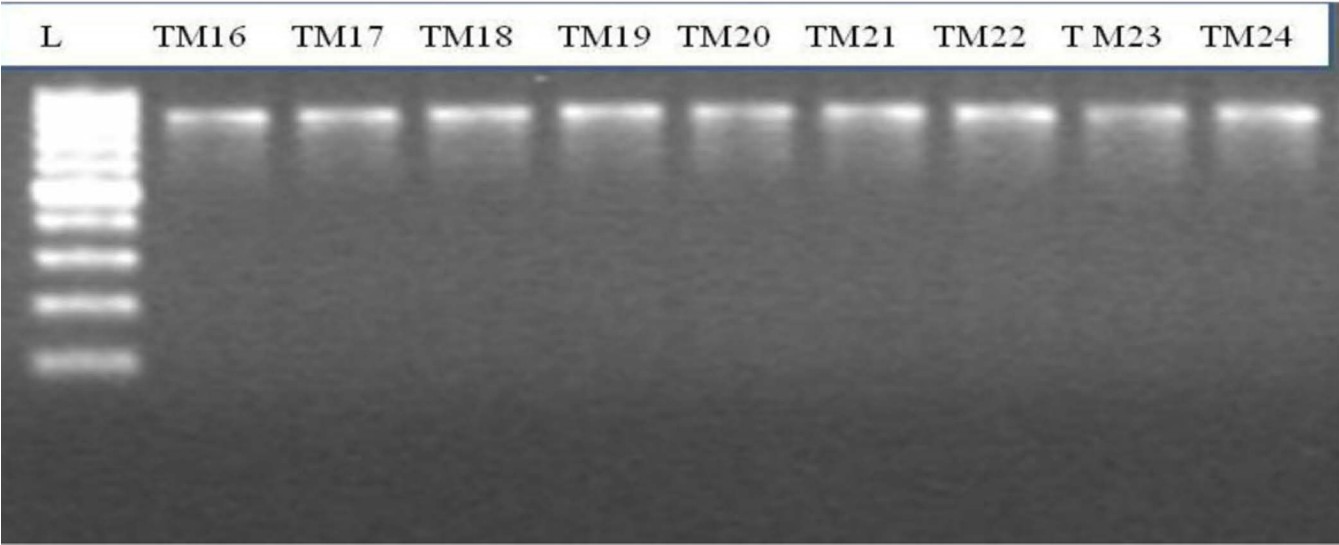

**Fig 1. 16s rRNA amplification of isolates (TM1-TM24) of size ~1500 bp and L is 1Kb Ladder.**

valerate produced by the isolates ranged from 50.37 mM to 83.59 mM, 6.92 mM to 17.87 mM, 1.10 mM to 2.77 mM, 3.08 mM to 10.34 mM, 0.70 mM to 1.56 mM and 0.39 mM to 1.61 mM, respectively. Isolate TM6 produced highest (P < 0.01) acteate (83.59 mM), propionate (17.87 mM), iso-bytyrate (2.77 mM), butyrate (10.34 mM), iso-valerate (1.56 mM) and valerate (1.61 mM) compared to other isolates. Lowest SCFAs (64.19 mM) was produced by isolate TM7. The isolates TM5 and TM11 produced the lowest amount of propionate and butyrate. The acetate to propionate ratio (C2:C3) was significantly higher in isolate TM5 and TM11.

## Discussion

The diverse symbiotic bacterial species found in termite guts contributes to the breakdown of lignocellulose, oligosaccharides, and aromatic compounds, suggesting that these bacteria

**Table 5. 16S rDNA sequencing based identification.**

| Isolates | Nearest Valid Taxon | % similarity | Sequence base pairs | Phylum | Family | Gene Bank Accession |
|----------|---------------------|--------------|---------------------|--------|--------|---------------------|
| TM1 | *Bacillus megaterium strain IAM 13418* | 99 | 1473 | *Firmicutes* | *Bacillaceae* | MT356127 |
| TM2 | *Citrobacter freundii strain JCM 1657* | 98 | 1417 | *Proteobacteria* | *Enterobacteriaceae* | MT356128 |
| TM3 | *Citrobacter farmeri strain CDC 2991-81* | 99 | 1472 | *Proteobacteria* | *Enterobacteriaceae* | MT356129 |
| TM4 | *Citrobacter freundii strain JCM 1657* | 98 | 1427 | *Proteobacteria* | *Enterobacteriaceae* | MT356130 |
| TM5 | *Bacillus flexus strain IFO15715* | 99 | 1530 | *Firmicutes* | *Bacillaceae* | MT356131 |
| TM6 | *Pilibacter termitis strain TI-1* | 98 | 1413 | *Firmicutes* | *Enterococcaceae* | MT356132 |
| TM7 | *Bacillus megaterium strain IAM* | 99 | 1482 | *Firmicutes* | *Bacillaceae* | MT356133 |
| TM8 | *Bacillus licheniformis strainATCC14580* | 99 | 1485 | *Firmicutes* | *Bacillaceae* | MT356134 |
| TM9 | *Bacillus cereus strain ATCC 14579* | 98 | 1524 | *Firmicutes* | *Bacillaceae* | MT356135 |
| TM10 | *Clostridium termitidis* | 99 | 1446 | *Firmicutes* | *Hungateiclostridiaceae* | MT356136 |
| TM11 | *Clostridium saccharobutylicum strain P262* | 100 | 1353 | *Firmicutes* | *Clostridiaceae* | MT356137 |
| TM12 | *Clostridium indolis strain 7* | 97 | 1414 | *Firmicutes* | *Lachnospiraceae* | MT356138 |
| TM13 | *Clostridium termitidis* | 98 | 1446 | *Firmicutes* | *Hungateiclostridiaceae* | MT356139 |
| TM14 | *Bacillus oleronius strain DSM 9356* | 98 | 1477 | *Firmicutes* | *Bacillaceae* | MT356140 |
| TM15 | *Clostridium cellulovorans 743B* | 96 | 1447 | *Firmicutes* | *Clostridiaceae* | MT356141 |
| TM16 | *Lactococcus nasutitermitis strain M19* | 97 | 1446 | *Firmicutes* | *Streptococcaceae* | MT356142 |
| TM17 | *Clostridium sporogenes strain JCM 1416* | 90 | 1535 | *Firmicutes* | *Clostridiaceae* | MT356143 |
| TM18 | *Lactococcus nasutitermitis strain M19* | 99 | 1440 | *Firmicutes* | *Streptococcaceae* | MT356144 |
| TM19 | *Bacillus circulans strain ATCC 4513* | 98 | 1482 | *Firmicutes* | *Bacillaceae* | MT356145 |
| TM20 | *Bacillus circulans strain ATCC 4513* | 98 | 1482 | *Firmicutes* | *Bacillaceae* | MT356146 |
| TM21 | *Ruminiclostridium papyrosolvens DSM 2782* | 98 | 1382 | *Firmicutes* | *Hungateiclostridiaceae* | MT356147 |
| TM22 | *Brevibacillus brevis strain DSM 30* | 96 | 1447 | *Firmicutes* | *Paenibacillaceae* | MT356148 |
| TM23 | *Clostridium puniceum strain BL 70/20* | 96 | 1481 | *Firmicutes* | *Clostridiaceae* | MT356149 |
| TM24 | *Clostridium sporogenes strain JCM 1416* | 99 | 1489 | *Firmicutes* | *Clostridiaceae* | MT356150 |

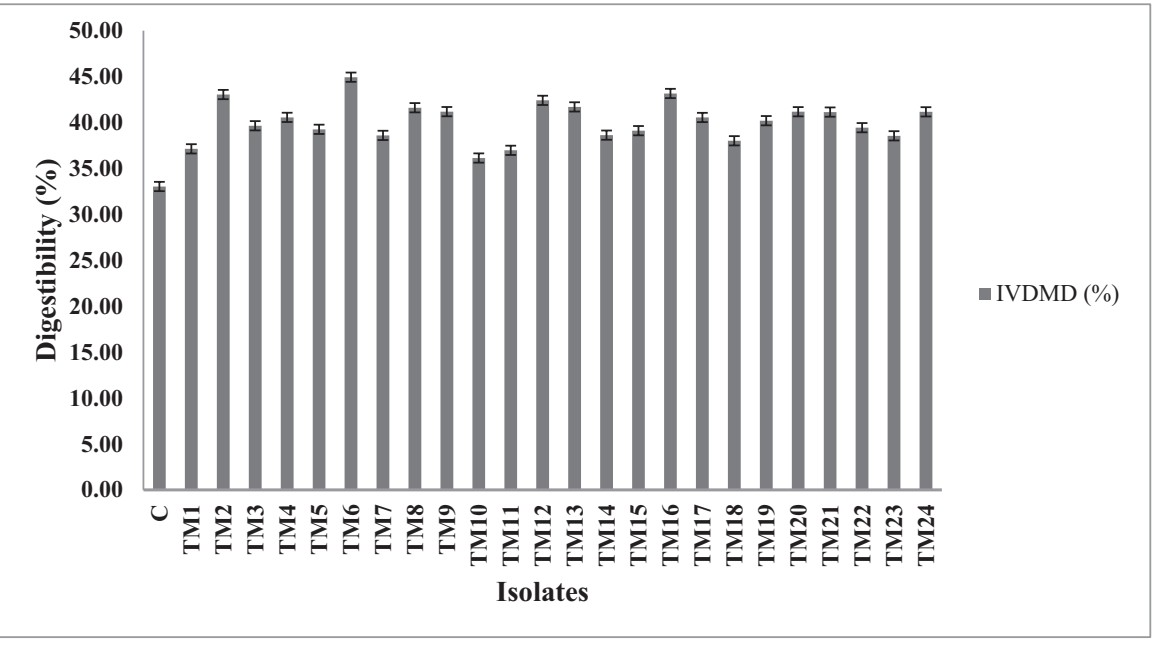

**Fig 2. Effect of inclusion of bacterial isolates on in vitro dry matter (IVDMD) digestibility at 48 h post incubation.**

**Table 6. Fermentation attributes total short chain fatty acid production.**

| Isolates | Acetate(C2) mM | Propionate (C3) mM | Iso-Butyrate (C4i) mM | Butyrate (C4) mM | Iso-Valerate (C5i) mM | Valerate (C5) mM | Total SCFAs mM | BcFA | NGR | C3:(C2⁺C4) | C2:C3 |
|---|---|---|---|---|---|---|---|---|---|---|---|
| TM1 | 56.74[i] | 14.26[f] | 1.71[bcd] | 7.34[g] | 0.96[def] | 1.18[g] | 82.19[hij] | 3.85[cde] | 4.16[ab] | 0.22[defg] | 3.98[abc] |
| TM2 | 70.72[k] | 14.09[ef] | 2.83[f] | 6.93[defg] | 1.15[fg] | 1.33[g] | 97.05[k] | 5.32[fg] | 4.64[bc] | 0.18[b] | 5.03 cd |
| TM3 | 60.88[j] | 14.10[ef] | 1.96[cde] | 7.30[fg] | 0.92[def] | 0.98[f] | 86.15[j] | 3.87[cde] | 4.42[abc] | 0.21 cd | 4.32[abc] |
| TM4 | 56.96[i] | 13.96[ef] | 1.86[bcd] | 7.16[fg] | 1.01[efg] | 1.27[g] | 82.23[hij] | 4.14[e] | 4.18[ab] | 0.22[def] | 4.09[abc] |
| TM5 | 61.35[j] | 6.93[a] | 2.82[f] | 3.08[ab] | 0.96[def] | 0.39[b] | 75.54[efg] | 4.17[e] | 6.55[e] | 0.11[a] | 8.97[f] |
| TM6 | 83.59[l] | 17.87[g] | 2.77[f] | 10.34[h] | 1.56[h] | 1.61[h] | 117.73[l] | 5.93[g] | 4.63[bc] | 0.19[bc] | 4.68[bcd] |
| TM7 | 44.82[b] | 10.61[b] | 1.37[bcd] | 5.92[def] | 0.73[bcd] | 0.74[cde] | 64.19[b] | 2.84[bc] | 4.43[abc] | 0.21 cd | 4.23[abc] |
| TM8 | 51.47[efg] | 13.43[def] | 1.54[bcd] | 7.15[fg] | 0.93[def] | 0.96[f] | 75.50[efg] | 3.44[bcde] | 4.11[ab] | 0.23[defh] | 3.84[ab] |
| TM9 | 49.26[de] | 13.43[def] | 1.42[bcd] | 7.03[defg] | 0.80[bcde] | 0.97[f] | 72.91[cdef] | 3.19[bcde] | 4.01[ab] | 0.24[efghi] | 3.67[ab] |
| TM10 | 44.82[b] | 10.61[b] | 1.37[bcd] | 5.92[def] | 0.73[bcd] | 0.74[cde] | 64.19[b] | 2.84[bc] | 4.43[abc] | 0.21 cd | 4.23[abc] |
| TM11 | 55.17[hi] | 6.92[a] | 2.60[e] | 3.67[bc] | 1.23[g] | 0.56[c] | 70.15[bcde] | 4.39[ef] | 5.96[de] | 0.12[a] | 8.05[e] |
| TM12 | 63.17[j] | 11.29[bc] | 2.05[de] | 4.46[c] | 1.13[fg] | 0.90[ef] | 83.00[ij] | 4.08[de] | 4.97[c] | 0.17[b] | 5.61[d] |
| TM13 | 51.44[efg] | 13.14[cdef] | 1.53[bcd] | 6.74[defg] | 0.82[bcde] | 0.97[f] | 74.65[defg] | 3.32[bcde] | 4.15[ab] | 0.23[defg] | 3.92[ab] |
| TM14 | 44.26[b] | 12.10[bcde] | 1.37[bcd] | 6.14[defg] | 0.71[bcd] | 0.79[def] | 65.39[b] | 2.88[bcd] | 3.98[ab] | 0.24[fghi] | 3.67[ab] |
| TM15 | 50.37[ef] | 14.85[f] | 1.44[bcd] | 7.41[g] | 0.90[cdef] | 0.98[f] | 75.96[efgh] | 3.32[bcde] | 3.78[a] | 0.26[i] | 3.40[a] |
| TM16 | 55.47[hi] | 14.10[ef] | 1.54[bcd] | 7.15[fg] | 0.93[def] | 0.96[f] | 80.16[ghij] | 3.44[bcde] | 4.20[abc] | 0.23[defg] | 3.96[abc] |
| TM17 | 45.67[bc] | 13.23[cdef] | 1.30[abcd] | 6.55[defg] | 0.71[bcd] | 0.75[cde] | 68.20[bcd] | 2.76[bc] | 3.86[ab] | 0.25[hi] | 3.46[a] |
| TM18 | 48.24[cde] | 11.22[b] | 1.17[abc] | 5.74[de] | 0.65[bc] | 0.62 cd | 67.63[bc] | 2.44[b] | 4.55[abc] | 0.21 cd | 4.30[abc] |
| TM19 | 49.25[de] | 13.89[ef] | 1.20[abc] | 6.65[defg] | 0.71[bcd] | 0.78[def] | 72.48[cdef] | 2.70[bc] | 3.93[ab] | 0.25[ghi] | 3.55[a] |
| TM20 | 49.96[def] | 13.34[def] | 1.10[ab] | 6.91[defg] | 0.73[bcd] | 0.81[def] | 72.85[cdef] | 2.64[bc] | 4.16[ab] | 0.23[deghi] | 3.75[ab] |
| TM21 | 50.66[efg] | 14.02[ef] | 1.18[abc] | 6.91[defg] | 0.78[bcde] | 0.76[cde] | 74.30[defg] | 2.71[bc] | 4.02[ab] | 0.24[fghi] | 3.62[ab] |
| TM22 | 46.86[bcd] | 11.75[bcd] | 1.20[abc] | 5.68[d] | 0.63[b] | 0.65 cd | 66.77[bc] | 2.48[b] | 4.27[abc] | 0.22[defg] | 3.99[abc] |
| TM23 | 53.80[ghi] | 14.54[f] | 1.21[abc] | 7.09[efg] | 0.75[bcde] | 0.79[def] | 78.19[fghi] | 2.76[bc] | 4.09[ab] | 0.24[efghi] | 3.70[ab] |
| TM24 | 53.12[fgh] | 13.22[cdef] | 1.29[abcd] | 6.93[defg] | 0.70[bcd] | 0.81[def] | 76.07[efgh] | 2.79b[c] | 4.36[abc] | 0.22[def] | 4.02[abc] |
| Control | 24.63[a] | 5.66[a] | 0.58[a] | 2.04[a] | 0.15[a] | 0.19[a] | 33.26[a] | 0.92[a] | 4.50[abc] | 0.21[cde] | 4.35[abc] |
| SEM | 1.21 | 0.318 | 0.071 | 0.05 | 0.03 | 0.03 | 1.64 | 0.12 | 0.07 | 0.03 | 0.155 |

Branch – chain fatty acids (BcFA) = valerate + iso − valerate + iso − butyrate (C5 + C5i + C4i).NGR Non - glucogenic: Glucogenicratio- = [(Acetate + 2 × Butyrate + BcFA ÷ Propionate + BcFA-} or [C2 + 2C4 + C5 + C5i + C4i- ÷ C3 + C5 + C5i + C4i-}.a,b,c Means bearing different superscript differ significantly (p < 0.05). SCFA = short-chain fatty acids, SEM = standard error of mean.

are important for the digestion of fiber [31, 32]. Till date, several natural polymer-degrading microbes have been isolated and characterized from the termite gut for degradation of plant biomass to biofuel. However, the present study aimed identification of potentially useful anaerobic fibre degrading bacteria from the gut of wood feeding lower termites inhabiting in the semi-arid environments so that these isolates could be used to improve the usage of fibrous plant biomass in ruminants' feeding.

The isolates showed diverse colony characteristics in terms of size, shape and colour. Fourteen isolates were gram-positive rods, while, remaining 8 were gram negative rods and 2 were gram positive cocci. The presence of cellulolytic bacilli and cocci in different species of termites is not uncommon [33–36]. The, isolates were also biochemically characterized and illustrated using Bergey's manual of determinative bacteriology following morphological screening [37]. In consistent with the present results, several studies have reported similar biochemical and carbohydrate utilization characteristics of bacterial isolates from the gut of termites *Reticulitermes santonensis* [38], *Zootermopsis angusticollis* [39], *Anacanthotermes vagans* [40] and *Microcerotermes diversus* [13].

Termites degrade fibre either by their own enzymatic machinery or from enzymes of symbiotic microbes residing in their gut. The association between termites and microbial symbionts in gut makes it an efficient source of enzymes like cellulase, hemicellulase and auxillary enzymes [31]. Similar to the present findings, a prior study found that bacteria isolated from the gut of higher termite (*Nasutitermes* sp.) showed endoglucanase activity ranging from 0.5 to 6.8 U/ml [41]. Furthermore, *Bacillus* sp was also isolated from wood eating termites *Neotermes* spp and *Cryptotermes brevis* exhibiting endoglucanase activity of 0.29 and 1.78 U/ml, respectively [35,42]. Avicel was utilized as a substrate for exoglucanase activity since the cell walls of plants are made of microcrystalline cellulose that is interlaced with biopolymers [43]. In line of the present findings, an exoglucanase activity of 7.93 to 18.76 U/ml was reported in bacteria isolated from hindgut of wood eating termite *Amitermes evuncifer silvestri* [34]. A similar FPase activity of 16.18 U/ml was reported in bacteria isolated from the gut of termite *Odontotermes formosanus* [44, 45]. However, in the present study bacteria isolated from the gut of *C. heimi* have higher exoglucanase activity than endoglucanase activity, which depicts that these isolates are better degraders of recalcitrant plant fiber that contains high crystalline cellulose. The β-glucosidase enzyme is found in the digestive system of many insects and lower termites, i.e., mostly present in the salivary gland of termites rather than their gut and helps in their energy metabolism [46]. However, bacteria isolated from the gut of higher termite *Macrotermes annandalei* revealed presence of β-glucosidase genes that may be involved in the host's fiber digestion [47]. Similarly, *Bacillus* spp. isolated from termite gut had β-glucosidase activity varying between 0.6 and 1.5 U/ml [48]. In lower termites, 81% of amylase activity was detected in salivary glands [49], however, microbial origin amylases were also found in the gut of termites produced by the symbiotic bacteria inhabiting in the gut [50]. Several studies have reported that higher termites have gut microbial population which produces amylase and it agrees with the present findings [46,51–53]. Degradation of hemicelluloses requires enzyme endo-1, 4-xylanase and studies have reported community of xylanolytic bacteria in the gut of termites [54]. Similar to our findings, bacteria belonging to the *Bacillus* group with promising xylanolytic activity have also been isolated from wood eating termite *Cryptotermes brevis* [35].

Filter paper contains both amorphous and crystalline cellulose. Degradation of filter paper by any bacteria indicates its cellulolytic potential. Previous studies also revealed filter paper degradation potential of bacterial species isolated from termites, however, their filter paper degradation potential was lower as compared to the present findings [55, 56]. The exoglucanase enzyme is mainly responsible for degradation of crystalline cellulose and it is directly related to filter paper degradation potential of the isolates. Thus, an increased filter paper degrading potential of the isolates in comparison to previous reports signify their importance with high exoglucanase activity and could be promising inoculant for commercial enzyme production aimed at plant fiber degradation.

The evolutionary study revealed that 87% of the isolates belonged to the phyla *Firmicutes* and remaining to *Proteobacteria* (Fig 3). The majority of isolates from these two phyla belonged to the orders *Bacillales*, *Lactobacillales*, *Enterobacterales*, and *Clostridiales*. Isolates under the order *Clostridiales* belonged to the families *Hungaticlostridiaceae, Clostridiaceae* and *Lachnospiraceae*. Besides the order *Clostridiales,* some isolates also belonged to order *Bacillales, Lactobacillales* and *Enterobacterales* under the family *Bacillaceae, Streptococcaceae, Enterococcaceae* and *Enterobacteriaceae*, respectively (Fig 4). In the present study, isolates were mostly from *Clostridiaceae* family followed by *Bacillaceae, Enterobacteriaceae, Streptococcaceae* and *Enterococcaceae*. Earlier reports also observed similar grouping of isolates in the phylum *Firmicutes* and order *Clostridiales* [57]. The 16S rRNA gene-based identification (Fig 5) also revealed that *Fimicutes* phyla and *Clostridia* clustar was the most abundant bacterial

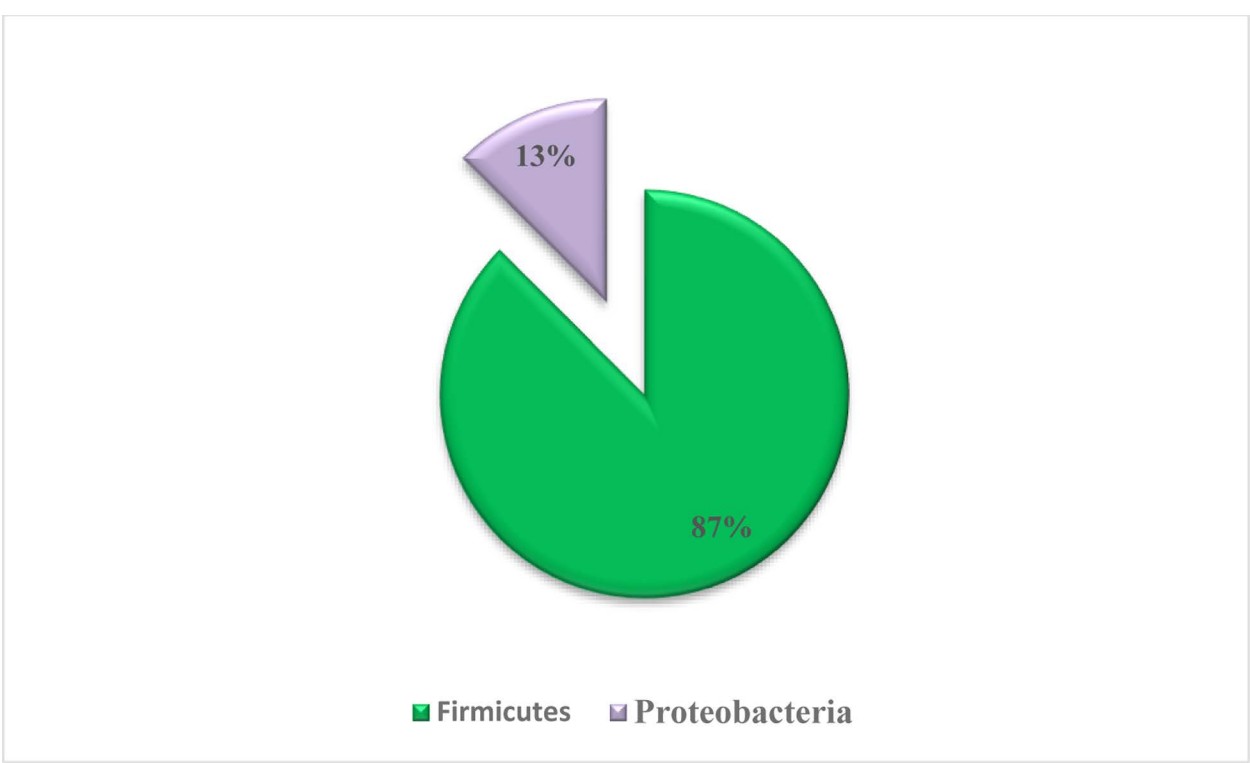

**Fig 3. Main phyla abundance of fibrolytic bacteria isolated from gut of wood eating lower termite (*Coptotermes heimi*).**

population including major representatives of *Peptococcaceae, Gracilibacteraceae, Clostridiaceae* and *Ruminococcaceae family*. In a previous study *Bacteroidetes*, *Fibrobacteres* and *Proteobacteria* were also reported to be present in the hindgut of wood-eating termites [58]. The *Clostridales* group contained members of the *Clostridiaceae* family, which is important for the breakdown of plant biomass and indicates an isolate's capacity for cellulolysis [58]. Majority of the bacterial populations in the mid gut of termite *Nasutitermes corniger* belongs to the phylum *Firmicutes* and are members of the family *Lachnospiraceae* which are capable of breaking down cellulose and hemicelluloses [59]. The genetic library of termites fed on xylan revealed *Firmicutes* as the most prevalent phylum, and this suggests that *Firmicutes* related bacteria are able to degrade hemicelluloses [60]. Moreover, it was also reported that a high abundance of *Enterobacteriales* was found in the foregut of termites (*Bullitermes* sp.) after 16S rRNA metagenomic investigation of the community structures of bacteria in the gut [58].

In the present study, inoculation of screened isolates enhanced nutrient digestibility of the substrate containing *Cenchrus ciliaris* and *Vigna mungo* straw. It is reported that microbial enzymes which break down structural fibers increases the amount of sugars available to rumen bacteria, which enhances nutrient digestibility [61]. Several studies have reported a symbiotic relationship between facultative and obligate anaerobes in the termite gut, where, facultative anaerobes scavenge oxygen that penetrates from gut exoskeleton which is essential for fibrolytic activity of obligatory anaerobes [62, 63]. The isolate TM6 had highest endo and exoglucanase activity which resulted in enhanced DM digestibility. Likewise, a study reported improvement in nutrient digestibility of substrate containing wheat straw and date leaves by inoculation of symbiotic bacteria isolated from the termite gut [13]. However, no effect was observed on *in vitro* rumen fermentation and fibre degradation when lignocellulose-degrading

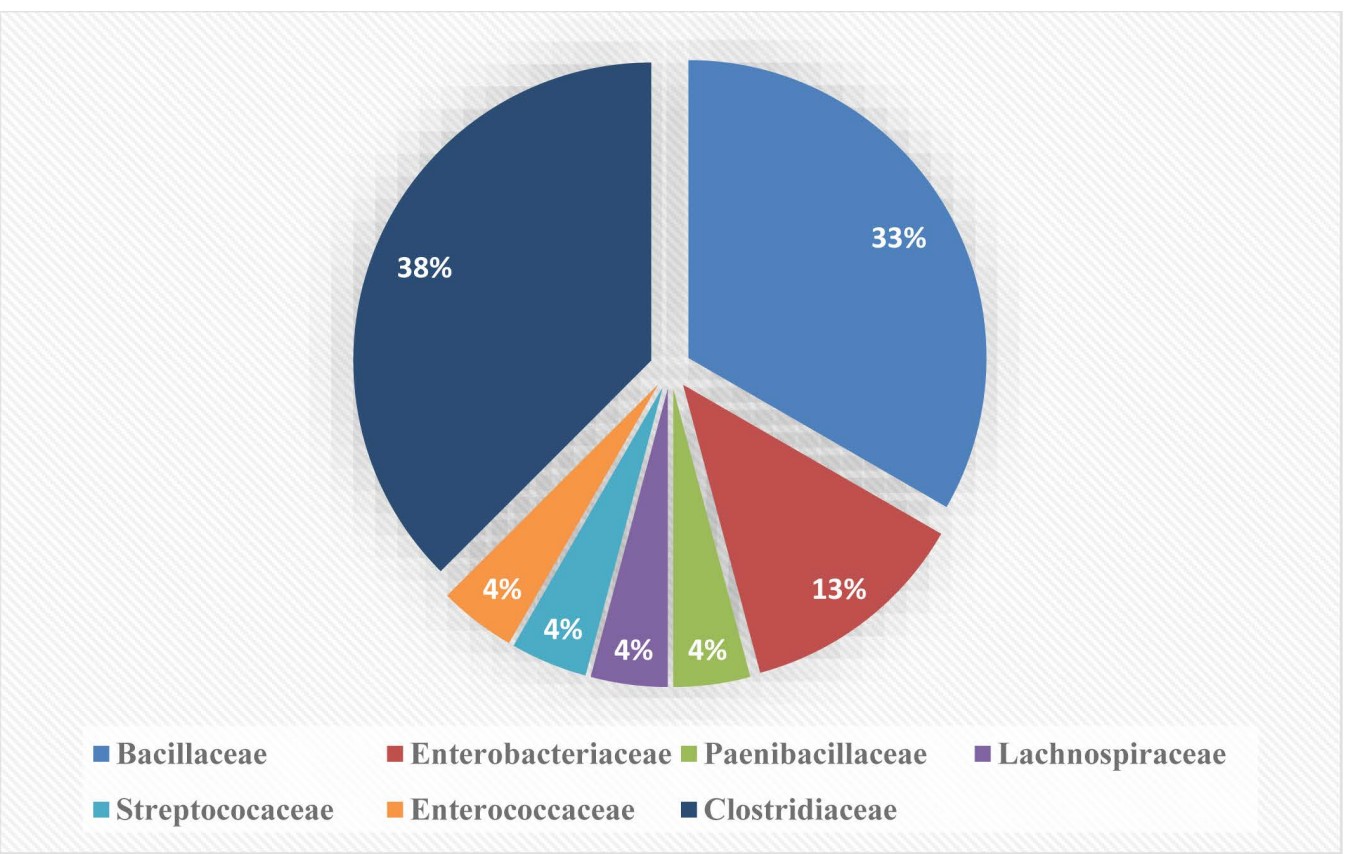

**Fig 4. Distribution of fibrolytic bacteria identified in the gut of wood eating lower termite (*Coptotermes heimi*) at family level.**

bacteria isolated from termite gut was innoculated with sheep rumen fluid [63]. The plant fiber containing ligno-cellulosic and hemicellulosic complexes needs to be broken down for releasing the carbohydrate moiety as energy source and carbon-skeleton for the multiplication of rumen microbial population that aid in nutrient availability to ruminants for increased productivity [15]. Thus, promising bacterial isolates from the termite gut with enhanced fiber-degrading activity could be a potential source for *in vitro* and/or *in vivo* manipulation to maximize fibrous-feed utilization for animal production.

## Conclusions

Till date, this is the first study regarding the existence of isolates with varying fibrolytic enzyme activity in the gut of the wood-eating termite *Coptotermes heimi*. Phylogenetic analysis depicted four major families, i.e., *Clostridiales, Bacillales, Lactobacillales* and *Enterobacterales* under phyla *Firmicutes* and *Proteobacteria* with the ability to produce cellulolytic and xylanolytic enzymes. Furthermore, inoculation of isolated bacterial strains enhanced digestibility of roughage-based diet up to 36% and maximum improvement was observed in case of TM6. Although, further studies are required to understand the interaction of these isolates with the rumen microbiota if they are used as direct fed microbial for improving degradation of poor quality roughages in ruminants. Hence, *in vivo* studies are required to validate the effect of these exogenous isolates in rumen environment for confirming their efficacy to convert low nutritive biomass as quality ruminant feed. Nonetheless, the identified isolates could

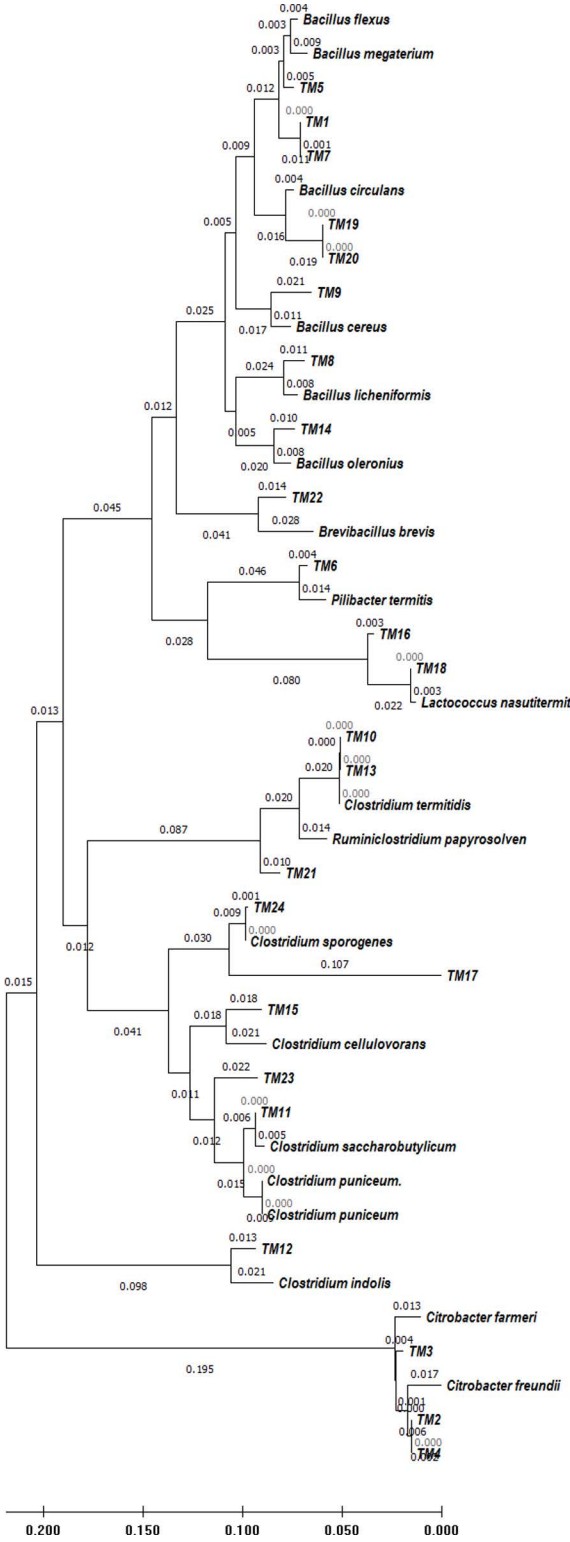

**Fig 5. Phylogentic tree based on the 16S rRNA partial sequences of the isolates.** Scale bar = 1 inferred nucleotide substitution per 50 nucleotides.

serve as promising inoculants in commercial enzyme production aimed at *in situ* plant-fiber degradation.

## Supporting information

**S1 File. Raw images.**
(DOCX)

## Acknowledgments

Authors are thankful to the Vice Chancellor, Mewar University, Chittorgarh and Director, ICAR-Central Sheep and Wool Research Institute for providing necessary research facilities required during this study.

## Author contributions

**Conceptualization:** Artabandhu Sahoo.

**Data curation:** Pankaj Kumar Kumawat, Srobana Sarkar.

**Formal analysis:** Pankaj Kumar Kumawat, Srobana Sarkar, Satish Kumar.

**Funding acquisition:** Artabandhu Sahoo.

**Investigation:** Pankaj Kumar Kumawat.

**Methodology:** Pankaj Kumar Kumawat.

**Project administration:** Artabandhu Sahoo.

**Resources:** Artabandhu Sahoo.

**Software:** Srobana Sarkar, Satish Kumar.

**Supervision:** Satish Kumar, Artabandhu Sahoo.

**Visualization:** Srobana Sarkar.

**Writing – original draft:** Pankaj Kumar Kumawat, Srobana Sarkar.

**Writing – review & editing:** Srobana Sarkar.

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
