## [Decision Letter · Decision Letter 0]

24 Mar 2024

PONE-D-24-05915Lower termite (Coptotermes heimi) gut fibrolytic bacterial consortium: Isolation, fiber degradation potential, phylogenetic dynamics and invitro digestibilityPLOS ONE

Dear Dr. Sahoo,

Thank you for submitting your manuscript to PLOS ONE. After careful consideration, we feel that it has merit but does not fully meet PLOS ONE’s publication criteria as it currently stands. Therefore, we invite you to submit a revised version of the manuscript that addresses the points raised during the review process.

We look forward to receiving your revised manuscript.

Kind regards,

Vishal Ahuja

Academic Editor

PLOS ONE

Journal Requirements:

Additional Editor Comments:

Dear authors,

The work presented in manuscript may have some significance but the representation is too poor. The writing part need major revision.

The detailed comments are given below

Line 14: Please provide the full name of the author.

Line 15: spacing before "Principal".

Line 19:change to "The present study was aimed to ..."

LIne 20: missing a spacing - of anaerobic; from gut; termite Coptotermes; inhabiting under; semi arid.

Line 21: missing a spacing before "A total".

Line 22: missing a spacing before "isolates".

Line 23: activity.

Line 24: The highest........the lowest...

Line 25: missing a spacing before "phylogenetic".

Line 26: missing a spacing before "In vitro"; between "substrate was".

Line 27: change to "TM6 revealed the highest".

Line 28: missing a spacing - the gut of C. heimi.

Line 30: missing a spacing before "utilization".

Line 33: to improve.

Line 33: poor quality of fibres.

Line 34: missing a spacing before Coptotermes, gut.

Line 37: into.

Line 40: higher and lower? incomplete sentence.

Line 41:harbour.

Line 41: no comma before and after "while".

Line 41: contain.

Line 43: work, help.

Line 43: breaking down complex.

Line 48: poor-quality. Delete "adding".

Line 50: have the ability.

Line 51: missing a spacing before "Furthermore".

Line 52: missing a comma between acetogenesis and methanogenesis. rewrite the last sentence.

Line 54: typo error - heimi.

Line 55: C. heimi.

Line 56: C. heimi.

Line 56: harbour a diverse...

Line 58: from the gut.

Line 59: of C. heimi.

Line 59: delete microbiota never studied before

Line 60: roughage-based

Line 65: C. heimi.

Line 66: what type of nest? What are the criteria in the site/tree selection?

Line 67: delete "trees". Replace "on March" with "in March". How did you collect the samples?

Line 80: 48 h.

Line 82: was repeated.....

Line 93: The experiment was conducted to estimate the enzymatic activity instead of the enzyme itself. Please rephrase the sentence.

Line 96: activities are expressed.

Line 104-105: rewrite.

Line 107: 39°C.

Line 110: rewrite.

Line 111-112: rewrite.

Line 113-116: rewrite.

Line 123: 10 d.

Line 125: was measured.

Line 128: Get the correct name of the kit.

Line 133: missing a spacing between a numeric and a unit.

Line 134: How many different types of conditions?

Line 135: 72ºC

Line 136: 1.5 Kb

Line 145: isolates were processed.....replicates...

Line 147: How many types of content?

Line 150: each bottle........condition.

Line 152: 100 mL.

Line 154: 24 h.

Line 156: italic - in vitro. the samples were strained.....

Line 157: -20°C

Line 160: missing a spacing - 6 ft.

Line 160: SP-1200 and 1% phosphoric acid.

Line 162: 150°C

Line 163: define gases air.

Line 164: missing a spacing - CFAs mixture, 65 mM. Small letter for acetic acid.

Line 173: missing a spacing before "All".

Line 174: missing a spacing - their ability.

Line 175: missing a spacing - which confirms.

Line 176: missing a spacing before "All".

Line 182: rewrite the sentence.

Line 185: The endoglucanase activity of TM23 (Clostridium puniceum strain BL 70/20) was ...

Line 186: missing a spacing - and the highest. Delete "was showed by isolate TM23.

Line 186-197: only the scientific names are written in italic. The coding names should be written in a regular format.

Line 191: The maximum activity was observed in TM18...

Line 198: missing a spacing - degradation potential.

Line 199: hours or h. Please standardise.

Line 201: The coding names should be written in a regular format.

Line 204; missing a spacing - 1500 bp. Explain why only 20 isolates were processed molecular identification.

Line 205: missing a spacing - to nucleotide blast.

Line 206-217: The coding names should be written in a regular format.

Line 217: 24 sequences represent how many bacterial isolates?

Line 221: no italic for "hay". missing a spacing - straw by.

Line 222: The coding names should be written in a regular format.

Line 225 - 226: the highest, the lowest.

Line 238 - 243: check all the spacing between words. A lot of grammatical errors. Rewrite the last sentence.

Line 250-251: spacing.

Line 252-259: no italic for sp. or spa.

Line 259-262: rewrite.

Line 263: from the gut of C. heimi.

Line 267: from the gut of .

Line 271-272: rewrite.

Line 272: to extract.

Line 295: the most......populations which include...

Line 296: delete "family".

Line 297: in the handgun of.

Line 298: a key role.

Line 299: in the fermentation processes.

Line 302: in the midgut of.

Line 304: Firmicutes as the most.

Line 305: it was reported.

Line 307: small letter - revealed.

Line 308: had enhanced the.

Line 309: spacing between texts.

Line 310: the highest.

Line 319: from the gut.

Line 320: this is the first report.

Line 321: the gut of C. heimi.

Good luck.

Reviewers' comments:

Reviewer's Responses to Questions

**Comments to the Author**

1. Is the manuscript technically sound, and do the data support the conclusions?

Reviewer #1: Yes

2. Has the statistical analysis been performed appropriately and rigorously? 

Reviewer #1: Yes

3. Have the authors made all data underlying the findings in their manuscript fully available?

Reviewer #1: Yes

4. Is the manuscript presented in an intelligible fashion and written in standard English?

Reviewer #1: No

5. Review Comments to the Author

Reviewer #1: Line 14: Please provide the full name of the author.

Line 15: spacing before "Principal".

Line 19:change to "The present study was aimed to ..."

LIne 20: missing a spacing - of anaerobic; from gut; termite Coptotermes; inhabiting under; semi arid.

Line 21: missing a spacing before "A total".

Line 22: missing a spacing before "isolates".

Line 23: activity.

Line 24: The highest........the lowest...

Line 25: missing a spacing before "phylogenetic".

Line 26: missing a spacing before "In vitro"; between "substrate was".

Line 27: change to "TM6 revealed the highest".

Line 28: missing a spacing - the gut of C. heimi.

Line 30: missing a spacing before "utilization".

Line 33: to improve.

Line 33: poor quality of fibres.

Line 34: missing a spacing before Coptotermes, gut.

Line 37: into.

Line 40: higher and lower? incomplete sentence.

Line 41:harbour.

Line 41: no comma before and after "while".

Line 41: contain.

Line 43: work, help.

Line 43: breaking down complex.

Line 48: poor-quality. Delete "adding".

Line 50: have the ability.

Line 51: missing a spacing before "Furthermore".

Line 52: missing a comma between acetogenesis and methanogenesis. rewrite the last sentence.

Line 54: typo error - heimi.

Line 55: C. heimi.

Line 56: C. heimi.

Line 56: harbour a diverse...

Line 58: from the gut.

Line 59: of C. heimi.

Line 59: delete microbiota never studied before

Line 60: roughage-based

Line 65: C. heimi.

Line 66: what type of nest? What are the criteria in the site/tree selection?

Line 67: delete "trees". Replace "on March" with "in March". How did you collect the samples?

Line 80: 48 h.

Line 82: was repeated.....

Line 93: The experiment was conducted to estimate the enzymatic activity instead of the enzyme itself. Please rephrase the sentence.

Line 96: activities are expressed.

Line 104-105: rewrite.

Line 107: 39°C.

Line 110: rewrite.

Line 111-112: rewrite.

Line 113-116: rewrite.

Line 123: 10 d.

Line 125: was measured.

Line 128: Get the correct name of the kit.

Line 133: missing a spacing between a numeric and a unit.

Line 134: How many different types of conditions?

Line 135: 72ºC

Line 136: 1.5 Kb

Line 145: isolates were processed.....replicates...

Line 147: How many types of content?

Line 150: each bottle........condition.

Line 152: 100 mL.

Line 154: 24 h.

Line 156: italic - in vitro. the samples were strained.....

Line 157: -20°C

Line 160: missing a spacing - 6 ft.

Line 160: SP-1200 and 1% phosphoric acid.

Line 162: 150°C

Line 163: define gases air.

Line 164: missing a spacing - CFAs mixture, 65 mM. Small letter for acetic acid.

Line 173: missing a spacing before "All".

Line 174: missing a spacing - their ability.

Line 175: missing a spacing - which confirms.

Line 176: missing a spacing before "All".

Line 182: rewrite the sentence.

Line 185: The endoglucanase activity of TM23 (Clostridium puniceum strain BL 70/20) was ...

Line 186: missing a spacing - and the highest. Delete "was showed by isolate TM23.

Line 186-197: only the scientific names are written in italic. The coding names should be written in a regular format.

Line 191: The maximum activity was observed in TM18...

Line 198: missing a spacing - degradation potential.

Line 199: hours or h. Please standardise.

Line 201: The coding names should be written in a regular format.

Line 204; missing a spacing - 1500 bp. Explain why only 20 isolates were processed molecular identification.

Line 205: missing a spacing - to nucleotide blast.

Line 206-217: The coding names should be written in a regular format.

Line 217: 24 sequences represent how many bacterial isolates?

Line 221: no italic for "hay". missing a spacing - straw by.

Line 222: The coding names should be written in a regular format.

Line 225 - 226: the highest, the lowest.

Line 238 - 243: check all the spacing between words. A lot of grammatical errors. Rewrite the last sentence.

Line 250-251: spacing.

Line 252-259: no italic for sp. or spa.

Line 259-262: rewrite.

Line 263: from the gut of C. heimi.

Line 267: from the gut of .

Line 271-272: rewrite.

Line 272: to extract.

Line 295: the most......populations which include...

Line 296: delete "family".

Line 297: in the handgun of.

Line 298: a key role.

Line 299: in the fermentation processes.

Line 302: in the midgut of.

Line 304: Firmicutes as the most.

Line 305: it was reported.

Line 307: small letter - revealed.

Line 308: had enhanced the.

Line 309: spacing between texts.

Line 310: the highest.

Line 319: from the gut.

Line 320: this is the first report.

Line 321: the gut of C. heimi.

6. PLOS authors have the option to publish the peer review history of their article (what does this mean? ). If published, this will include your full peer review and any attached files.

**Do you want your identity to be public for this peer review?** For information about this choice, including consent withdrawal, please see our Privacy Policy .

Reviewer #1: **Yes: ** Wei Hong Lau

---

## [Author Response · Author response to Decision Letter 1]

31 May 2024

Dear Editor,

Thank you for your useful suggestions on the structure of our manuscript. The suggestions were helpful in improving our manuscript and we have addressed them in the revised version.

Here we are also providing the detailed responses to the comments and a word file has also been attached

Comments Response to comments

Line 14: Please provide the full name of the author Agreed and complied

Line 15: spacing before "Principal". Corrected

Line 19: change to "The present study was aimed to..." Sentence modified as suggested

LIne 20: missing a spacing - of anaerobic; from gut; termite Coptotermes; inhabiting under; semi arid. Needful corrections done

Line 21: missing a spacing before "A total". Needful corrections done

Line 22: missing a spacing before "isolates". Needful corrections done

Line 23: activity. Corrected as suggested

Line 24: The highest........the lowest... Sentence modified as suggested

Line 25: missing a spacing before "phylogenetic". Needful corrections done

Line 26: missing a spacing before "In vitro"; between "substrate was". Needful corrections done

Line 27: change to "TM6 revealed the highest". Needful corrections done

Line 28: missing a spacing - the gut of C. heimi. Needful corrections done

Line 30: missing a spacing before "utilization". Needful corrections done

Line 33: to improve. Needful corrections done

Line 33: poor quality of fibres. Needful corrections done

Line 34: missing a spacing before Coptotermes, gut. Needful corrections done

Line 37: into. Corrected as suggested

Line 40: higher and lower? incomplete sentence. Sentence completed

Line 41:harbour. Corrected as suggested

Line 41: no comma before and after "while". Needful corrections done

Line 41: contain.. Corrected as suggested

Line 43: work, help. Corrected as suggested

Line 43: breaking down complex Corrected as suggested

Line 48: poor-quality. Delete "adding". Corrected as suggested

Line 50: have the ability. Corrected as suggested

Line 51: missing a spacing before "Furthermore". Corrected as suggested

Line 52: missing a comma between acetogenesis and methanogenesis. rewrite the last sentence.

Line 54: typo error - heimi. Sentence rewritten as suggested

Line 55: C. heimi. Needful corrections done

Line 56: C. heimi. Needful corrections done

Line 56: harbour a diverse. Corrected as suggested

Line 58: from the gut. Corrected as suggested

Line 59: of C. heimi. Corrected as suggested

Line 59: delete microbiota never studied before Corrected as suggested

Line 60: roughage-based Corrected as suggested

Line 65: C. heimi. Corrected as suggested

Line 66: what type of nest? What are the criteria in the site/tree selection? The Coptotermes is part of family Rhinotermitidae are construct their nests within the wood

they consume. These termites carve out intricate networks of tunnels within the wood, devoid

of any visible external entrances, except for temporary ones formed during swarming periods.

The tunnels within the nests are divided by partitions composed of their waste material and

are often covered or lined with a plaster-like substance also made from their waste.

The criteria used to identifying termite infested trees are as followed

1. Visible Damage

2. Frass or Termite Droppings

3. Mud Tubes

4. Wood Density and Moisture Content

Line 67: delete "trees". Replace "on March" with "in March". How did you collect the samples? Corrected as suggested

Line 80: 48 h. Corrected as suggested

Line 82: was repeated..... Corrected as suggested

Line 93: The experiment was conducted to estimate the enzymatic activity instead of the enzyme itself. Please rephrase the sentence. Corrected as suggested

Line 96: activities are expressed. Corrected as suggested

Line 104-105: rewrite. Sentence rewritten as suggested

Line 107: 39°C. Corrected as suggested

Line 110: rewrite. Sentence rewritten as suggested

Line 111-112: rewrite. Sentence rewritten as suggested

Line 113-116: rewrite. Sentence rewritten as suggested

Line 123: 10 d. Corrected as suggested

Line 125: was measured. Corrected as suggested

Line 128: Get the correct name of the kit. Needful corrections done

Line 133: missing a spacing between a numeric and a unit. Corrected as suggested

Line 134: How many different types of conditions? The conditions are mentioned in text and rewritten for clarity.

Line 135: 72ºC Needful corrections done

Line 136: 1.5 Kb Needful corrections done

Line 145: isolates were processed.....replicates... Text rewritten for clarity.

Line 147: How many types of content? Text rewritten for clarity.

Line 150: each bottle........condition. Needful corrections done

Line 152: 100 mL. Needful corrections done

Line 154: 24 h. Needful corrections done

Line 156: italic - in vitro. the samples were strained..... Needful corrections done

Line 157: -20°C Needful corrections done

Line 160: missing a spacing - 6 ft. Needful corrections done

Line 160: SP-1200 and 1% phosphoric acid. Needful corrections done

Line 162: 150°C Needful corrections done

Line 163: define gases air. Needful corrections done

Line 164: missing a spacing - CFAs mixture, 65 mM. Small letter for acetic acid. Corrected as suggested

Line 173: missing a spacing before "All". Corrected as suggested

Line 174: missing a spacing - their ability. Corrected as suggested

Line 175: missing a spacing - which confirms.

Line 176: missing a spacing before "All". Corrected as suggested

Line 182: rewrite the sentence. Sentence rewritten for clarity

Line 185: The endoglucanase activity of TM23 (Clostridium puniceum strain BL 70/20) was ... Corrected as suggested

Line 186: missing a spacing - and the highest. Delete "was showed by isolate TM23. Corrected as suggested

Line 186-197: only the scientific names are written in italic. The coding names should be written in a regular format. Agreed and complied

Line 191: The maximum activity was observed in TM18... Corrected as suggested

Line 198: missing a spacing - degradation potential. Corrected as suggested

Line 199: hours or h. Please standardise. Agreed and complied

Line 201: The coding names should be written in a regular format. Agreed and complied

Line 204; missing a spacing - 1500 bp. Explain why only 20 isolates were processed molecular identification. Corrected as suggested. All 24 isolates were processed for molecular identification, 20 was typographical error.

Line 205: missing a spacing - to nucleotide blast. Needful corrections done

Line 206-217: The coding names should be written in a regular format. Agreed and complied

Line 217: 24 sequences represent how many bacterial isolates? Sequences of 24 isolates were submitted to NCBI GenBank

Line 221: no italic for "hay". missing a spacing - straw by. Agreed and complied

Line 222: The coding names should be written in a regular format. Agreed and complied

Line 225 - 226: the highest, the lowest. Needful corrections done

Line 238 - 243: check all the spacing between words. A lot of grammatical errors. Rewrite the last sentence. Sentence rewritten for clarity

Line 250-251: spacing. Needful corrections done

Line 252-259: no italic for sp. or spa. Needful corrections done

Line 259-262: rewrite. Sentence rewritten for clarity

Line 263: from the gut of C. heimi. Corrected as suggested

Line 267: from the gut of . Corrected as suggested

Line 271-272: rewrite. Sentence rewritten

Line 272: to extract. Corrected as suggested

Line 295: the most......populations which include.. Corrected as suggested

Line 296: delete "family". Corrected as suggested

Line 297: in the handgun of. Corrected as suggested

Line 298: a key role. Corrected as suggested

Line 299: in the fermentation processes. Corrected as suggested

Line 302: in the midgut of. Needful corrections done

Line 304: Firmicutes as the most. Corrected as suggested

Line 305: it was reported. Corrected as suggested

Line 307: small letter - revealed. Corrected as suggested

Line 308: had enhanced the. Corrected as suggested

Line 309: spacing between texts. Needful corrections done

Line 310: the highest. Needful corrections done

Line 319: from the gut. Corrected as suggested

Line 320: this is the first report. Needful corrections done

Line 321: the gut of C. heimi Needful corrections done

Regards

Artabandhu Sahoo

---

## [Decision Letter · Decision Letter 1]

18 Jul 2024

PONE-D-24-05915R1Lower termite ( Coptotermes heimi ) gut fibrolytic bacterial consortium: Isolation, phylogenetic characterization, fibre degradation potential and in vitro digestibilityPLOS ONE

Dear Dr. Sahoo,

Thank you for submitting your manuscript to PLOS ONE. After careful consideration, we feel that it has merit but does not fully meet PLOS ONE’s publication criteria as it currently stands. Therefore, we invite you to submit a revised version of the manuscript that addresses the points raised during the review process.

We look forward to receiving your revised manuscript.

Kind regards,

Vishal Ahuja

Academic Editor

PLOS ONE

Journal Requirements:

Additional Editor Comments:

Dear author,

Thank you for the submission. Review of your manuscript has been completed and it need minor revision. A lot of mistakes and type errors have been found in it which need to be removed for further consideration.

Thanks and regards.

Reviewers' comments:

Reviewer's Responses to Questions

**Comments to the Author**

1. If the authors have adequately addressed your comments raised in a previous round of review and you feel that this manuscript is now acceptable for publication, you may indicate that here to bypass the “Comments to the Author” section, enter your conflict of interest statement in the “Confidential to Editor” section, and submit your "Accept" recommendation.

Reviewer #1: (No Response)

Reviewer #2: (No Response)

2. Is the manuscript technically sound, and do the data support the conclusions?

Reviewer #1: Yes

Reviewer #2: Yes

3. Has the statistical analysis been performed appropriately and rigorously? 

Reviewer #1: Yes

Reviewer #2: I Don't Know

4. Have the authors made all data underlying the findings in their manuscript fully available?

Reviewer #1: Yes

Reviewer #2: Yes

5. Is the manuscript presented in an intelligible fashion and written in standard English?

Reviewer #1: Yes

Reviewer #2: Yes

6. Review Comments to the Author

Reviewer #1: Line 15: Please provide the full name of the author.

Line 40: Delete comma

Line 172: Double full stops.

Reviewer #2: Comments:

1. Line 23: The term ‘isolates’ is repeated after (TM1 to TM24). Please correct it.

2. Line 24: You mentioned that all isolates were obligatory anaerobes except four (TM8, TM9, TM14 and TM22). Kindly mention whether these four isolates are facultative anaerobes or aerobes.

3. Line 48: There is no spacing between fiber[15]. Please correct it.

4. Line 55: Double full stop after abundance. Please correct it.

5. Line 74: Whether it is Pfenning trace elements or Pfennig trace elements. Kindly confirm.

6. Line 82: -80°C (minus 80°C) should be written combinedly.

7. Line 153: There is no spacing between gfor. Please correct it.

8. Line 173: Double full stop. Please correct it.

9. Line 183: There is no spacing between (12.05 U/ml)and. Kindly correct it.

10. Line 186: There is no spacing between TM5and. Kindly correct it.

11. Line 283: Double full stop. Please correct.

12. Line 286: Whether it is termites fed xylan or termites fed on xylan. Kindly correct it.

13. Kindly recheck the manuscript for spacing and punctuation mistakes.

7. PLOS authors have the option to publish the peer review history of their article (what does this mean? ). If published, this will include your full peer review and any attached files.

**Do you want your identity to be public for this peer review?** For information about this choice, including consent withdrawal, please see our Privacy Policy .

Reviewer #1: No

Reviewer #2: No

---

## [Author Response · Author response to Decision Letter 2]

24 Jul 2024

Dear Editor,

Thank you and the reviewers for the useful suggestions on the structure of our manuscript. The suggestions were helpful in improving our manuscript and we have addressed them in the revised version.

Here we are also providing the detailed responses to the comments

Comments Response to comments

Reviewer #1

Line 15: Please provide the full name of the author. Needful corrections made

Line 40: Delete comma Corrected as suggested

Line 172: Double full stops Corrected as suggested

Reviewer #2

Line 23: The term ‘isolates’ is repeated after (TM1 to TM24). Please correct it. Needful corrections made

Line 24: You mentioned that all isolates were obligatory anaerobes except four (TM8, TM9, TM14 and TM22). Kindly mention whether these four isolates are facultative anaerobes or aerobes. Sentence modified as suggested

Line 48: There is no spacing between fiber. Please correct it. Needful corrections made

Line 55: Double full stop after abundance. Please correct it. Needful corrections made

Line 74: Whether it is Pfenning trace elements or Pfennig trace elements. Kindly confirm. Needful corrections made

Line 82: -80°C (minus 80°C) should be written combinedly. Corrected as suggested

Line 153: There is no spacing between g for. Please correct it. Corrected as suggested

Line 173: Double full stop. Please correct it. Corrected as suggested

Line 183: There is no spacing between (12.05 U/ml) and. Kindly correct it. Needful corrections made

Line 186: There is no spacing between TM5and. Kindly correct it. Needful corrections made

Line 283: Double full stop. Please correct. Corrected as suggested

Line 286: Whether it is termites fed xylan or termites fed on xylan. Kindly correct it. Corrected as suggested

Kindly recheck the manuscript for spacing and punctuation mistakes. Agreed and complied

---

## [Decision Letter · Decision Letter 2]

6 Aug 2024

PONE-D-24-05915R2Lower termite ( Coptotermes heimi ) gut fibrolytic bacterial consortium: Isolation, phylogenetic characterization, fibre degradation potential and in vitro digestibilityPLOS ONE

Dear Dr. Sahoo,

Thank you for submitting your manuscript to PLOS ONE. After careful consideration, we feel that it has merit but does not fully meet PLOS ONE’s publication criteria as it currently stands. Therefore, we invite you to submit a revised version of the manuscript that addresses the points raised during the review process.

We look forward to receiving your revised manuscript.

Kind regards,

Vishal Ahuja

Academic Editor

PLOS ONE

Journal Requirements:

**Additional Editor Comments:**

Dear authors,

Thank you for the submission. The reviewer asked for some minor details about methodology.

Kindly address the comments.

Thanks and regards.

Reviewers' comments:

Reviewer's Responses to Questions

**Comments to the Author**

1. If the authors have adequately addressed your comments raised in a previous round of review and you feel that this manuscript is now acceptable for publication, you may indicate that here to bypass the “Comments to the Author” section, enter your conflict of interest statement in the “Confidential to Editor” section, and submit your "Accept" recommendation.

Reviewer #3: All comments have been addressed

2. Is the manuscript technically sound, and do the data support the conclusions?

Reviewer #3: Yes

3. Has the statistical analysis been performed appropriately and rigorously? 

Reviewer #3: Yes

4. Have the authors made all data underlying the findings in their manuscript fully available?

Reviewer #3: Yes

5. Is the manuscript presented in an intelligible fashion and written in standard English?

Reviewer #3: Yes

6. Review Comments to the Author

Reviewer #3: (No Response)

7. PLOS authors have the option to publish the peer review history of their article (what does this mean? ). If published, this will include your full peer review and any attached files.

**Do you want your identity to be public for this peer review?** For information about this choice, including consent withdrawal, please see our Privacy Policy .

Reviewer #3: No

---

## [Editor Report · Decision Letter 3]

7 Jan 2025

PONE-D-24-05915R3Lower termite ( Coptotermes heimi ) gut fibrolytic bacterial consortium: Isolation, phylogenetic characterization, fibre degradation potential and in vitro digestibilityPLOS ONE

Dear Dr. Sahoo,

Thank you for submitting your manuscript to PLOS ONE. After careful consideration, we feel that it has merit but does not fully meet PLOS ONE’s publication criteria as it currently stands. Therefore, we invite you to submit a revised version of the manuscript that addresses the points raised during the review process.

We look forward to receiving your revised manuscript.

Kind regards,

Vishal Ahuja

Academic Editor

PLOS ONE

Journal Requirements:

Additional Editor Comments:

Dear authors,

Thanks for the submission of your revised manuscript. However, no response fille received. Kindly share it with clear response to al the comments.

Good luck.

---

## [Author Response · Author response to Decision Letter 4]

8 Jan 2025

Dear Editor,

Thank you and the reviewers for the useful suggestions on the structure of our manuscript. The suggestions were helpful in improving our manuscript and we have addressed them in the revised version.

Herewith we are also providing the detailed responses to the comments and it is also attached as a separate file

Response to comments:

1. Query - Could author explain in further detail the rationale behind using particularly Coptotermes heimi for this study: Answer - Wood-eating termites possess remarkable potential in various biotechnological applications due to their ability to efficiently break down recalcitrant biomass. Coptotermes heimi are the most predominant lower termites the in semi-arid regions of India and there exists similarity between the environments in the rumen and termites' gut. This unique capability makes them valuable source of fiber-degrading enzymes which could be utilized to enhance fiber digestion in ruminants, improving feed efficiency and overall productivity in livestock systems.

2. Query - Line 157-158, authors should provide the concentration of samples used for analysis with GC: Answer - Needful corrections made as suggested

3. Query - In Table 3, based on trends and value of fibrolytic enzyme activites observed, could author comment on finding some interesting correlation between activity and isolates: Answer - Agreed and added in the discussion

4. Query - In Table 4, could author comment on providing some quantitative/numerical data for filter paper degradation potential. Also, please provide remarks on significance of “+”in context of value: Answer - This is a qualitative test hence, no numerical data is available. Although, the significance of ‘+’ is added as a footnote below the Table.

Regards

Artabandhu Sahoo

---

## [Editor Report · Decision Letter 4]

10 Jan 2025

Lower termite ( Coptotermes heimi ) gut fibrolytic bacterial consortium: Isolation, phylogenetic characterization, fibre degradation potential and in vitro digestibility

PONE-D-24-05915R4

Dear Dr. Sahoo,

We’re pleased to inform you that your manuscript has been judged scientifically suitable for publication and will be formally accepted for publication once it meets all outstanding technical requirements.

Kind regards,

Vishal Ahuja

Academic Editor

PLOS ONE

Additional Editor Comments (optional):

Dear authors,

Thanks for the submission of revised version. The manuscript has been revised suitably for consideration.

Good luck
---

## [Editor Report · Acceptance letter]

PONE-D-24-05915R4

PLOS ONE

Dear Dr. Sahoo,

I'm pleased to inform you that your manuscript has been deemed suitable for publication in PLOS ONE. Congratulations! Your manuscript is now being handed over to our production team.

Kind regards,

on behalf of

Dr. Vishal Ahuja

Academic Editor

PLOS ONE